# Efficacy of Noninvasive Brain Stimulation (tDCS or TMS) Paired with Language Therapy in the Treatment of Primary Progressive Aphasia: An Exploratory Meta-Analysis

**DOI:** 10.3390/brainsci10090597

**Published:** 2020-08-28

**Authors:** Nicole R. Nissim, Paul J. Moberg, Roy H. Hamilton

**Affiliations:** 1Laboratory for Cognition and Neural Stimulation, Department of Neurology, University of Pennsylvania, Philadelphia, PA 19104, USA; Nicole.Nissim@pennmedicine.upenn.edu; 2Moss Rehabilitation Research Institute, Elkins Park, PA 19027, USA; 3Department of Psychiatry, University of Pennsylvania, Philadelphia, PA 19104, USA; Moberg@pennmedicine.upenn.edu; 4Department of Otorhinolaryngology: Head & Neck Surgery, University of Pennsylvania, Philadelphia, PA 19104, USA; 5Department of Neurology, University of Pennsylvania, Philadelphia, PA 19104, USA

**Keywords:** noninvasive brain stimulation (NIBS), transcranial direct current stimulation (tDCS), transcranial magnetic stimulation (TMS), repetitive TMS, neurostimulation, primary progressive aphasia, behavioral language therapy, language function, intervention

## Abstract

Noninvasive brain stimulation techniques, such as transcranial direct current stimulation (tDCS) and transcranial magnetic stimulation (TMS), paired with behavioral language therapy, have demonstrated the capacity to enhance language abilities in primary progressive aphasia (PPA), a debilitating degenerative neurological syndrome that leads to declines in communication abilities. The aim of this meta-analysis is to systematically evaluate the efficacy of tDCS and TMS in improving language outcomes in PPA, explore the magnitude of effects between stimulation modalities, and examine potential moderators that may influence treatment effects. Standard mean differences for change in performance from baseline to post-stimulation on language-related tasks were evaluated. Six tDCS studies and two repetitive TMS studies met inclusion criteria and provided 22 effects in the analysis. Random effect models revealed a significant, heterogeneous, and moderate effect size for tDCS and TMS in the enhancement of language outcomes. Findings demonstrate that naming ability significantly improves due to brain stimulation, an effect found to be largely driven by tDCS. Future randomized controlled trials are needed to determine long-term effectiveness of noninvasive brain stimulation techniques on language abilities, further delineate the efficacy of tDCS and TMS, and identify optimal parameters to enable the greatest gains for persons with PPA.

## 1. Introduction

Advances in neuroscience, neuroimaging, and neurorehabilitation have expanded our understanding of the important functional role of neuroplasticity in the recovery or preservation of neurologic function in brain disorders [1,2]. Correspondingly, research has grown in the field of neuromodulation, including noninvasive brain stimulation (NIBS) techniques, which aims to alter neuroplasticity and enhance cognition to ultimately improve behavior [3,4]. Transcranial direct current stimulation (tDCS) and transcranial magnetic stimulation (TMS) are two widely used methods of noninvasive neuromodulation that offer safe and painless approaches to modulate neuroplasticity [5,6,7,8]. TMS is currently being employed in clinical practice to treat depression [9,10] and other psychiatric conditions, and translational and clinical research exploring both forms of brain stimulation has expanded because these NIBS approaches demonstrate their potential as therapeutic tools for a variety of neurological syndromes [11,12,13,14,15]. While tDCS and TMS differ in a number of ways—from their underlying mechanism(s) of action and impact on the brain to the cost and ease of application—both technologies can be used to enhance or inhibit cortical excitability [16]. Studies show that repeated application of NIBS (e.g., multiple sessions) can induce persistent changes in cognition and behavior [16,17]. In recent years, these techniques have been explored as adjunctive therapies to treat neurological syndromes that affect language processing, including aphasia in the setting of strokes, and more recently neurodegenerative language loss, termed primary progressive aphasia (PPA) [18,19].

PPA results from degeneration of brain regions within the language network. Three clinical variants have been identified that differ with respect to their pattern of phenotypic presentation, underlying neuropathology, and location/distribution of degeneration [20,21]. Nonfluent/agrammatic PPA (naPPA) is associated with left frontal lobe atrophy and associates with grammatical processing deficits in language production, speech apraxia, and labored or effortful speech [22]. Semantic variant PPA (svPPA) associates with left anterior and ventral temporal lobe atrophy and typically causes deficits in word comprehension and naming with intact speech production [23]. Lastly, logopenic variant PPA (lvPPA) is characterized by left temporal and parietal lobe atrophy and associates with word retrieval deficits, slow, halting spontaneous speech, and repetition difficulty [19,20,24]. As the biological basis of language deficits in PPA becomes clearer, there is increasing impetus to explore the use of neurally-focused, targeted interventions, such as NIBS technologies.

Behavioral language therapies are currently the standard of care for patients with PPA, and many studies that have investigated NIBS approaches in this patient population have employed neuromodulation as an adjunct to conventional speech language therapy [25,26,27,28]. Evidence suggests that behavioral therapies on their own can be modestly effective [23,29]. These interventions typically focus on a specific impairment (e.g., therapy geared toward deficits in object or action naming) or activity/participation-based treatments, which aim to improve ability to participate in desired activities and tasks [30]. For instance, studies suggest that a variety of language therapy methods can facilitate naming ability, including repeated practice of target picture naming [31], generative naming of categories under time constraints [32], and errorless learning [33]. Behavioral language therapy studies in PPA have shown that gains on language measures are largely restricted to the domain that received training, and certain approaches appear to slow the progression of language deficits within those domains [34,35]. However, evidence regarding maintenance and generalization of behavioral language therapy gains in PPA is inconsistent [23].

The concurrent application of NIBS with behavioral language therapies is well-aligned with what is currently known about the mechanisms of NIBS. Brain-induced stimulation effects often appear to be effort or state-dependent; the behavior that occurs during/near the time of stimulation directly influences brain stimulation outcomes [36,37]. Across a variety of cognitive domains, studies assessing tDCS [37,38] and TMS [39] have shown that the targeted brain system benefits most from stimulation when that system is being engaged by a relevant behavioral task during or near the time of stimulation [40]. However, in the context of aphasia, challenging the language system during active brain stimulation can lead to a selective facilitation of the underlying language network and potentially further enhance language abilities over behavioral therapies alone [41,42]. Prior studies pairing brain stimulation with language therapy in PPA patients have used varying stimulation sites. Regions of stimulation have included the left frontal/prefrontal cortex, dorsolateral prefrontal cortex (DLPFC), and the frontotemporal region of the left hemisphere to engage the brain regions that are thought to underlie the concurrent tasks employed (oral or written and spelling therapy, individualized speech therapy (focused on naming), and narration of wordless children’s books, respectively). Previous studies have shown involvement of the DLPFC in naming in healthy participants [43], whereas patients with lesions in frontal and frontotemporal regions have shown associations of decreased complexity in story narration [44]. It is important to note a gap in knowledge pertaining to the appropriate stimulation location in PPA patients and the ways in which the targeted brain region might differ between variants. Behavioral language therapy approaches often differ in technique or the language process being targeted, providing further complexity; the relationship between optimal stimulation locations for different forms of language therapy in the treatment of PPA is not yet well understood.

It is important to consider the basic mechanisms and differential impact of tDCS and TMS on neurophysiology and brain function to inform and optimize the therapeutic use of NIBS techniques in the treatment of PPA. Conventional tDCS functions by applying a diffuse, weak direct electrical current (typically 1–2 mA intensity) through electrodes placed on the scalp. As current flows from the anode to cathode electrode, tDCS alters the sub-threshold resting membrane potential of neurons and modulates cortical excitability, resulting in either up-regulation or down-regulation of neuronal activity during and after stimulation has ended, although it does not cause direct stimulation of action potentials [6,45,46]. In contrast, TMS entails acute, spatially focal generation of rapidly fluxing magnetic fields, which penetrate the skull and generate supra-threshold electrical currents that depolarize underlying cortical neurons [12,47]. The resulting excitation or inhibition effects on the brain from TMS are dependent on the frequency of stimulation; repetitive TMS (rTMS) applied at a high frequency (5–20 Hz) results in excitation, whereas low frequency rTMS (1 Hz) causes inhibition [48]. Given the substantive differences between tDCS and TMS, it is possible that there are differences in the efficacy of the two approaches with respect to treating language deficits in PPA, although such differences have not been directly explored to date.

In addition to mechanistic considerations, practical differences between tDCS and TMS are just as important to consider in the advancement of NIBS interventions for PPA. Not only does cost, ease of use, precision, and portability differ between these technologies, but the flexibility of pairing the stimulation approach with behavioral language therapy highly differs between tDCS and TMS [49]. Compared to TMS, tDCS is advantageous in all of these areas. Specifically, with regards to pairing NIBS with behavioral therapies, TMS is considerably more difficult to apply because participants are required to sit still and limit head movement during stimulation. However, an advantage of TMS is its high spatial resolution, which may be more relevant in research applications. Studies that require focal stimulation paired with behavioral therapy would benefit from TMS over tDCS.

One similarity shared by tDCS and TMS is that they both have many modifiable parameters, which can be applied in various ways to induce or inhibit cortical excitability and thus alter brain activity [4,48,50]. These include intensity, duration, target location, number of sessions, and size of tDCS electrode(s) or the TMS coil [8,51,52]. While this feature of NIBS has enabled its widespread use across a variety of syndromes (e.g., depression, pain, memory), the large parameter space of tDCS and TMS is also a source of considerable methodological heterogeneity in NIBS studies of PPA [53]. Taken together, the heterogeneity of the disorder itself, varied stimulation approaches, parameters employed, different behavioral language therapy approaches paired with stimulation, and small number of patients represented in most studies make characterizing either the overall impact of NIBS technologies or the effectiveness of either individual modality for treating language deficits in PPA challenging, necessitating systematic assessment.

As NIBS in PPA is a small but growing field, it is not yet known whether tDCS or TMS might be more effective in persons with PPA. Prior meta-analyses involving neuromodulation in PPA have focused only on tDCS in PPA and naming abilities as the only outcome measure [41,54]. This meta-analysis aims to assess the efficacy of tDCS and TMS as adjuncts to language therapy in improving communication abilities in PPA. Naming is a commonly tested language skill in PPA research. Not surprisingly, most of the effects in this meta-analysis are from studies in which naming was the outcome measure. However, we also include an effect for speech production and grammatical comprehension, in addition to effects from a study that assessed performance on the Stroop task (color-word interference) to examine naming during cognitive interference and executive function demands (categorized as naming during cognitive inhibition). Despite the low number studies and effects beyond naming, they were included to provide efficacy estimates and gain insight into whether neuromodulation and language therapy can be effective for other important linguistic measures or differentially impact language outcomes. In this meta-analysis, we examine all language measures in an overall omnibus analysis and separately assess language categories as moderator variables in sub-analyses (e.g., naming, speech production, grammatical comprehension, and naming during cognitive inhibition). We also explore the magnitude of effects of tDCS versus TMS in sub-analyses. We hypothesize that a greater therapeutic benefit would be driven by tDCS compared to TMS when paired with language therapy. Our rationale for making this prediction is that because tDCS induces sub-threshold stimulation, its effects may be preferentially driven by behaviorally mediated state-dependent activation of relevant networks than supra-threshold stimulation induced by TMS [55,56]. As such, we reasoned that tDCS might be more effective when paired with behavioral language therapies than TMS. We also explore other potential moderators that may impact treatment efficacy, such as stimulation parameters and subject demographics within the included studies. In addition, because prior evidence suggests that persons with aphasia may perform differently based on the category of the speech target they are attempting to produce, such as actions versus objects [54,57,58,59], we explore whether this is a relevant distinction in the NIBS literature on PPA. Finally, we examine available studies for publication bias, in order to evaluate the degree to which it may be influencing the apparent efficacy of NIBS approaches represented in the scientific literature.

## 2. Materials and Methods

This systematic meta-analysis was conducted and reported in accordance with the Preferred Reporting Items for Systematic Reviews and Meta-Analyses (PRISMA) guidelines [60].

### 2.1. Literature Search Strategy

One reviewer (N.N.) carried out literature searches to identify treatment studies involving tDCS or TMS in neurodegenerative PPA. Articles were identified through a computerized literature search using the following databases: PubMed, Embase, Cochrane Central Register of Controlled Trials, and clinicaltrials.gov. Search terms included “aphasia” OR “primary progressive aphasia” OR “language dysfunction” OR “language disorders” OR “neurodegenerative primary progressive aphasia” AND “transcranial magnetic stimulation” OR “repetitive TMS” OR “transcranial direct current stimulation”. The search was limited to articles written in English that included studies published between January 1960 and January 2020. This approach identified 118 potential articles from PubMed, Embase, Cochrane Central Register of Controlled trials, and 2 records identified from ClinicalTrials.gov. The PRISMA flow diagram, shown in Figure 1, displays the procedures for study identification. An additional thorough manual review of the articles was performed, as described below.

### 2.2. Eligibility: Inclusion and Exclusion Criteria

Inclusion criteria for eligibility required: (1) research articles that enrolled human subjects diagnosed with PPA (i.e., post-stroke aphasia studies were excluded); (2) studies that involved a behavioral treatment intervention regardless of therapy approach (such as computerized language therapy involving images or videos versus printed images; approaches that focused on naming actions or objects, narration of wordless picture books, oral and written picture naming-spelling therapy, category fluency, and the Stroop task) paired with tDCS or TMS; (3) studies that reported raw baseline and post-intervention scores or changes in accuracy on language-related tasks (e.g., naming, category fluency, speech production, grammatical comprehension, and writing); (4) studies with two or more participants. Studies were included irrespective of experimental design and consisted of within-subject crossover trials, between-subject, randomized controlled trials, case series, and an open-label pilot study with baseline and post-intervention time points. Articles were excluded based on the following criteria: (1) case studies of a single subject; (2) post-stroke aphasia studies; (3) review articles; (4) studies that lacked language assessments; (5) studies that included pharmacological or other additional interventions. Common reasons for article exclusion were duplication within the literature search, review articles, or limited statistical reporting (e.g., conference abstracts). 

### 2.3. Literature Selection and Data Extraction

Manuscript titles, abstracts, and full texts were independently screened by two of the authors (N.N., P.M.). Any disagreement during the selection process was resolved through discussion and consensus. Final selected studies are summarized in Table 1 and study demographics are shown in Table 2. The extracted data included author and publication year, type of trial design, sample size, participant demographics (age, sex, education), disease variant, mean time (in years) post-diagnosis, stimulation parameters (duration and number of sessions, intensity, hemisphere, anode/cathode location for tDCS, target brain region for TMS coil placement), and mean performance on language outcome measures at baseline and post-intervention. In all studies, language improvement was measured by comparing the number of correct stimuli (i.e., naming or number of correct units for writing/spelling) produced before and after the intervention in active and sham groups or from active baseline to post-intervention in the case of the open-label pilot study with no sham control. All language outcomes from tDCS and TMS studies were analyzed in an overall omnibus analysis. Sub-analyses were performed to assess naming performance and different forms of naming; this was guided by further classifying naming domains into action versus object and treated versus untreated stimuli.

### 2.4. Data Analyses

Analyses were conducted using the Comprehensive Meta-Analysis software (CMA version 3.0; www.meta-analysis.com). All analyses used a random effects model approach, which takes into account study variability (heterogeneity) due to differences in methods and sample characteristics, as well as the small number of studies included in the meta-analysis; this approach relies on the assumption that observed treatment effects can vary due to real differences in the effect within each study, as well as from sampling variability or chance [63]. The main outcome measure, the accuracy on language assessments, was defined as the mean rate of correct responses (e.g., naming stimuli or correct number of oral, written/spelling units). The computed outcomes (effect sizes) were calculated based on the mean difference scores between active versus sham groups from baseline to post-intervention. For studies with no sham arm, the effect size was calculated using the mean difference score from the active stimulation groups’ baseline to post-intervention; scores were standardized by calculating Cohen’s d, which is the difference between two raw means divided by the pooled standard deviation (SD). When the mean and SDs were not available, d was calculated from reported univariate *F*-tests, *t*-statistics, or *p*-values. Effect sizes were classified similar to previous conventions as small (*d* ≥ 0.20), medium (*d* ≥ 0.45), or large (*d* ≥ 0.80) [64]. Confidence intervals (CIs) and z-transformations of the effect size were used to determine if the statistical significance (*p* < 0.05) was reached. Significance was achieved for mean effects within the 95% CI that did not span zero; this criterion provides strong evidence that, on average, the treatment effect is beneficial [63]. To explore how much of the total variability could be attributed to heterogeneity among studies, the Cochran Q-statistic was used [65]. Cochran’s Q-statistic computes the sum of the squared deviations of each study’s estimate from the overall meta-analysis estimate. This test enabled determination of whether there were genuine differences underlying the results of the included studies (heterogeneity) or whether a variation in findings was due to chance alone (homogeneity) [66]. The following data assignments were used across the 22 examined effects: Paired groups (difference, *p*), Paired groups (*N*, *t*-value), Paired groups (means, *p*), Independent groups (means, *p*), Raw difference paired (SE).

To explore potential influential factors on treatment effects, subgroup moderator analyses were performed on categorical variables that could presumably affect variations between the included studies. Using the Q-statistic, the modality of stimulation (tDCS versus TMS) was assessed across all language measures. The language effects were explored, further comparing language tasks within the following categories: naming (18 effects), speech production (1 effect), grammatical comprehension (1 effect), and naming during cognitive inhibition (2 effects). The general naming category was also assessed for action versus object naming and treated versus untreated naming stimuli. Treated stimuli consisted of practiced items during the language therapy versus untreated items that were novel and unpracticed to determine if therapy gains can transfer to untrained stimuli. Meta-regression, an extension subgroup analysis that assesses moderator variables using regression-based techniques, was used to examine characteristics of continuous variables [67]. These variables included stimulation duration, number of treatment sessions, stimulation intensity (tDCS only), and participant demographics (age and sex). The two TMS studies included in this meta-analysis had comparable parameter settings and thus did not enable meaningful assessment of variation within the parameter space. 

Evaluation of publication bias was performed through graphic examination of the funnel plot, which provides a simple scatter plot of the intervention effect size estimates from each study plotted against the study precision or result. A relatively symmetrical inverted funnel shape indicates the absence of bias; an asymmetric funnel shape indicates that there is a systematic difference or bias between selected studies [68]. Egger’s regression provided a statistical measure quantifying funnel plot asymmetry [69]. Calculation of an adjusted rank-correlation test was also performed, according to the methods of Begg and Mazumdar [70]. Moreover, the classic fail-safe N [71] was used to identify the number of additional negative studies that would be needed to negate the current findings. 

## 3. Results

Of the total articles identified through the initial database search, 8 articles met our inclusion criteria and provided 22 effects in the meta-analysis. Two of the articles involved rTMS (*n* = 88 across effects), while six involved tDCS (*n* = 140 across effects) for an overall sample size of 115 subjects with PPA who underwent a form of behavioral language therapy paired with NIBS (tDCS *n* = 99; rTMS *n* = 16). Study details are shown in Table 2. 

### 3.1. Meta-Analysis: Treatment Effects across tDCS and TMS Studies

The omnibus analysis of overall treatment effects across tDCS and TMS modalities revealed that NIBS treatments resulted in a significant and moderate improvement in language functions over sham (*k* = 22, *d* = 0.46, 95% CI = 0.21–0.71; *z* = 3.61, *p* < 0.001). However, homogeneity analysis indicated that study-specific effect sizes were significantly heterogeneous (Q-stat = 51.6, *df* = 21, *p* < 0.001). Given that variability in study-specific effect sizes across tDCS and TMS treatment modalities differed more than would be expected from a sampling error alone, a moderator analysis was performed to better account for this heterogeneity. Study statistics and a corresponding forest plot are provided in Figure 2.

### 3.2. Moderator Analyses

#### Stimulation Modality: tDCS Versus TMS

Analysis of stimulation modality revealed that tDCS treatment (*k* = 10, *d* = 0.82, 95% CI = 0.44–1.19) yielded significantly larger improvements in language function relative to TMS treatment (*k* = 12, *d* = 0.17, 95% CI = −0.16–0.51) (Q-stat = 6.31, *df* = 1, *p* = 0.012).

### 3.3. Language Function 

Homogeneity analysis across the language categories (naming, speech production, grammatical comprehension, and naming during cognitive inhibition) revealed significant heterogeneity (Q-stat = 16.61, *df* = 3, *p* = 0.001) with speech production, showing significantly greater improvement with stimulation relative to both naming (Q-stat = 6.59, *df* = 1, *p* = 0.01) and naming during cognitive inhibition (Q-stat = 10.80, *df* = 1, *p* = 0.001). Grammatical comprehension showed significantly greater improvement with stimulation relative to naming during cognitive inhibition (Q-stat = 7.51 *df* = 1, *p* = 0.006) but was not significantly different relative to the general naming category (Q-stat = 2.88, *df* = 1, *p* = 0.08). A contrast of naming with naming during cognitive inhibition (Stroop color-word interference) also revealed a significant difference in the efficacy of stimulation (Q-stat = 5.98, *df* = 1, *p* = 0.014), with naming abilities showing greater improvement relative to naming during interference. 

### 3.4. Treated Versus Untreated Naming Stimuli

Contrasts between treated versus untreated naming did not reveal any significant differences (Q-stat = 0.44, *df* = 1, *p* = 0.51).

### 3.5. Stimulation Parameters: tDCS

#### 3.5.1. Intensity

Meta-regression for the sub-analysis comparing tDCS stimulus intensity (1.5 mA, *k* = 3 versus 2mA, *k* = 7) demonstrated a significant difference in intensity (Q-stat = 4.36; *df* = 1; *p* = 0.037) such that 1.5 mA (*d* = 1.56) was more beneficial than 2 mA (*d* = 0.59). 

#### 3.5.2. Stimulation Duration

Contrasts of 20 min stimulation duration (*k* = 8) versus 25 min stimulation duration (*k* = 2) did not reveal any significant differences (*z* = 0.77, *p* = 0.44). 

#### 3.5.3. Frequency of Treatment

The number of active treatment sessions was not a significant moderator for treatment effects (*p* > 0.05). 

### 3.6. Participant Demographics 

#### 3.6.1. Sex

Meta-regression revealed that studies with a higher percentage of male patients had an adverse impact on efficacy of stimulation, regardless of stimulation type (*z* = −2.61, *p* = 0.0089). 

#### 3.6.2. Age

The mean age of patients was 67.1 years. Analysis showed no significant moderation of effect size by age (*z* = −0.72, *p* = 0.47).

### 3.7. Publication Bias

Evaluation of publication bias revealed significant Begg (1-tailed *p* = 0.015) and Egger (1-tailed *p* = 0.049) tests, suggesting the possibility of bias in this sample of literature. Trim-and-fill analyses identified two putative outlier effects, which, if excluded, only minimally reduced the omnibus effect size (i.e., *d* = 0.39). Lastly, calculation of the classic fail-safe N revealed that 162 negative or “null” results would be needed to negate the present findings. Collectively, the latter findings suggest that the current meta-analytic results accurately represent the extant literature concerning tDCS and TMS stimulation as supplemental tools with behavioral language therapy in the treatment of PPA. Figure 3 displays the funnel plot for all included studies.

## 4. Discussion

This systematic meta-analysis explored the efficacy of pairing two NIBS approaches—tDCS and TMS—with behavioral language therapy to improve language outcomes in individuals with PPA. We found strong positive effects that support the efficacy of active tDCS and TMS over sham stimulation in significantly improving language abilities in PPA. Previous meta-analyses of this topic have only investigated the use of tDCS and have only employed oral or written naming as outcome measures. The current meta-analysis extended this literature by including TMS, which enabled an exploratory sub-analysis comparing the magnitude of effects between the neuromodulation modalities. These data suggest tDCS may be more efficacious than TMS as an adjunct to behavioral language therapy. Our findings are consistent with prior meta-analyses that demonstrate that tDCS is effective at improving oral and written naming in PPA patients. Our results did not demonstrate significant differences between action versus object naming or treated versus untreated stimuli. Moreover, in addition to naming assessments, we included grammatical comprehension and speech production outcomes in the overall analysis to assess the effects of NIBS on other language measures. Although there was only one effect for grammatical comprehension and speech production, these language output measures demonstrated the strongest effects compared to the naming effects, suggesting that these components of language can improve due to NIBS and language therapy. This indicates the necessity of exploring more diverse language measures, in addition to naming, in future PPA studies. Interestingly, exploratory moderator analyses of parameter settings revealed that lower intensity of tDCS (1.5 mA versus 2 mA) provides a greater benefit to language outcomes. Collectively, these data support the use of NIBS techniques paired with behavioral language therapy in the enhancement of language abilities in PPA.

The observation that tDCS paired with behavioral language therapies is more efficacious than TMS when paired with therapies aligned with our predictions. Our hypothesis was based on mechanistic differences between the two NIBS techniques. Because tDCS induces sub-threshold shifts in neuronal resting membrane potentials, its ability to elicit enduring effects on behavior are thought to rely upon and leverage Hebbian principles of neuroplasticity [72]. For this reason, it has been argued that tDCS ought to be most effective when paired with behavioral approaches that engage relevant networks [37,40,42], in this case the language network. Other differences between how tDCS and TMS are implemented may have also contributed to the observed difference in effect. tDCS pairs more easily with various forms of therapy due to its greater flexibility for movement and ease of use during the performance of a behavior. However, while it is generally easier to perform behavioral tasks and therapies concurrently with tDCS compared to TMS, the observed difference between tDCS and TMS in our analysis cannot be attributed to a systematic difference in whether the two technologies were administered “online” (i.e., concurrent with behavioral training) or “offline” (i.e., asynchronously with behavioral training), as all studies in our analysis delivered stimulation concurrent with behavioral tasks. Moreover, due to the fact that TMS provides focal stimulation, it requires structured hypotheses regarding the precise target region of stimulation. However, the question of which optimal brain regions are stimulated in persons with PPA remains unresolved. This may limit the efficacy of TMS in the treatment of PPA and is an aspect of this intervention that requires further exploration. One clear limitation in interpreting the observed difference between tDCS and TMS is that there were considerably fewer TMS studies in our sample, perhaps because tDCS is easier to administer than TMS in conjunction with behavioral therapies.

Naming is easily quantifiable and represents the most frequently tested language ability in aphasia research involving NIBS. Improvements in naming have repeatedly been shown after neuromodulation in persons with aphasia. Our moderator analyses confirmed that NIBS (both tDCS and TMS) can significantly and positively impact naming performance persons with PPA. Because evidence suggests that there may be important neural and behavioral differences in the ability to generate words within specific categories [54,57], we specifically examined whether there were any differences in action or object naming related to NIBS, but we found none. This lack of difference could be attributed to the low number of effects specifying different categories of naming. Surprisingly, the assessment of treated versus untreated naming stimuli showed no significant difference; we had expected treated stimuli to outperform untreated stimuli due to practice effects from training, learning, and memory. This finding may indicate that practice effects were not particularly strong in the included studies or that neuromodulation effects on the language performance may generalize across treated and untreated stimuli. While the moderator analysis, contrasting naming separately with grammatical comprehension, naming during cognitive inhibition, and speech production, indicated that the latter language measures demonstrated the strongest positive effects from active stimulation, these findings should be interpreted with caution due to low number of effects in speech production and the effects coming from a single open-label pilot study. However, these results do suggest that future research in PPA should include a wider array of language assessments that can enable greater insights into other language abilities beyond naming that might benefit from brain stimulation. 

In our examination of key stimulation parameters, the most noteworthy finding was that differences in stimulation intensity of tDCS may impact language outcomes and effects. Interestingly, our results favored lower intensity stimulation, demonstrated by the significant improvement in language outcomes with 1.5 mA compared to 2 mA intensity. This finding is broadly consistent with evidence that suggests that the relationship between the intensity of stimulation and its effects on physiology and behavior are complex and nonlinear [73]. On the other hand, this finding does not readily reconcile with recent evidence that suggests that individuals with greater brain atrophy may require higher intensities of stimulation to receive benefits [74,75,76] or with a small but growing body of evidence that suggests that higher intensities of tDCS (~4 mA in some studies) may be necessary to elicit consistent, robust effects on behavior, even in healthy individuals [77,78]. Unfortunately, within the available PPA literature, we were only able to compare two tDCS intensities of very similar magnitude (1.5 mA and 2 mA), limiting our ability to make definitive inferences. Analysis of differences in tDCS duration and number of treatment sessions in our meta-analysis did not reveal any significant findings and may relate to low variation in number of treatment sessions. Prior studies have shown that repeated stimulation sessions can elicit effects that outlast the period of stimulation [17,79]. These stimulation after-effects are thought to reflect long-term potentiation (LTP) mediated neuroplastic changes [80]. As tDCS research in PPA continues to move forward, it will remain critical to further clarify optimal stimulation parameters, such as intensity, duration, and frequency in the use of NIBS. 

Covariates, such as age, sex, education, time post-diagnosis, disease severity, and variant type, might influence NIBS effects on the brain. We were able to assess age, sex, and education and found a significant effect of sex, which indicated that females might benefit more than males from NIBS, irrespective of stimulation type. Differential effects of brain stimulation on gender have not been reported in PPA. This finding, amongst many other potential covariates that could not be readily assessed, requires further exploration. 

There are several limitations that should be considered when interpreting our findings. Although our search for appropriate studies was broad, our final number of studies for analysis was small. Differences in the behavioral treatments that were paired with neuromodulation comprised a potentially important source of heterogeneity in this study; the fact that these differences were not captured at a level of granularity to allow for direct investigation of the effects of different behavioral approaches was a limitation of this study that could be explored in future investigations. The number of tasks and effects outside of naming was limited, which points to the lack of diverse language measures in this body of literature and the need for other key language skills to be assessed in persons with PPA. Additionally, the sample sizes of these studies were low and there were fewer TMS studies than tDCS studies. This is to be expected to some extent, since the use of NIBS is relatively new in PPA. With only two TMS studies compared to six tDCS studies, it is difficult to reach a firm conclusion regarding efficacy between modality of stimulation. Although moderator analyses indicated that tDCS was more effective than TMS, this finding could be driven in part by a difference between the two in statistical power to identify an effect. Future research is needed to further elucidate whether one modality is more beneficial than the other in the treatment of PPA. 

Given that clinical characteristics and patterns of brain atrophy differ across PPA variants, disambiguating how brain stimulation techniques might affect each variant remains an important challenge. As work in this area matures, there may be a need for variant-specific, personalized stimulation approaches (e.g., stimulation targets being chosen based on subject-specific damage of the language network or a particular behavioral language therapy approach targeting the patient’s variant specific deficits). In our analysis, variant types differed across studies, which limited the ability to assess efficacy of NIBS on PPA variant types and language outcomes. Further research is needed to disentangle the impact of brain stimulation across these important dimensions, as well as across the spectrum of disease severity.

Finally, we were limited in our ability to assess some stimulation parameters, because investigators tended to select parameters within a fairly circumscribed range. Studies involving TMS had comparable parameters and did not permit further analysis. Similarly, the target location region of the anode and cathode in tDCS could not be meaningfully assessed, the hemisphere of stimulation could not be assessed in tDCS, and all behavioral language therapies occurred concurrent with stimulation, which did not permit the evaluation of offline tDCS or TMS effects. 

## 5. Conclusions

This meta-analysis has summarized the magnitude of effect of tDCS and TMS concurrent with behavioral language therapy in enhancing language abilities in persons with PPA within the current literature. We revealed a significant moderate effect of tDCS and TMS in improving language outcomes. Limited literature within the area of NIBS in PPAs did not enable meaningful assessment of all potential parameter influences of stimulation effects. In summary, tDCS and TMS evidence points toward a significant benefit in persons with PPA and holds promise as therapeutic treatment tools for this debilitating disease. Future research is needed to gain a greater understanding of the influence of parameters on stimulation outcomes and how to optimize these stimulation techniques to enable the greatest gains in the treatment of PPA.

## Figures and Tables

**Figure 1 brainsci-10-00597-f001:**
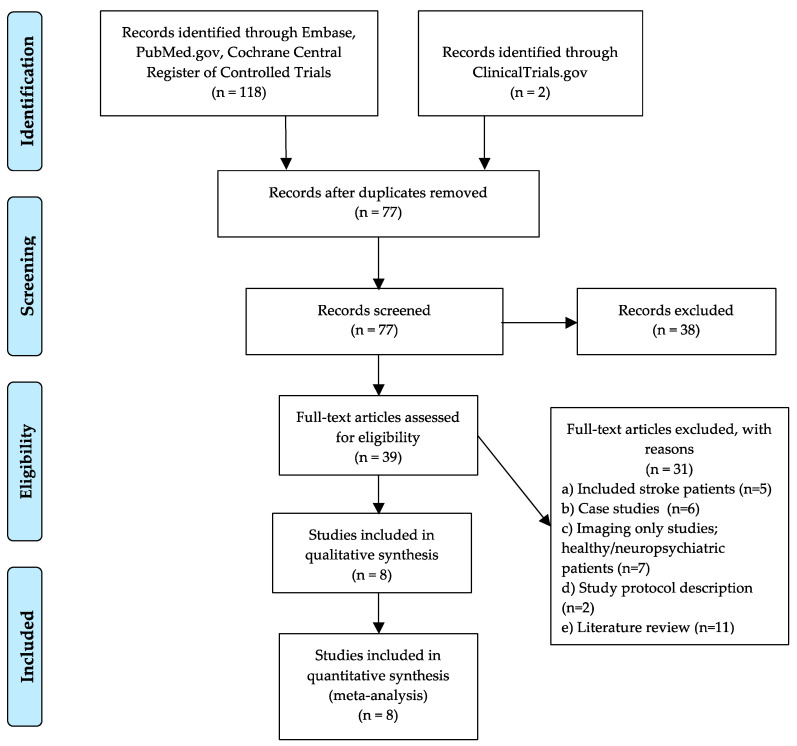
Preferred Reporting Items for Systematic Reviews and Meta-Analyses (PRISMA) flow diagram for the selection of studies.

**Figure 2 brainsci-10-00597-f002:**
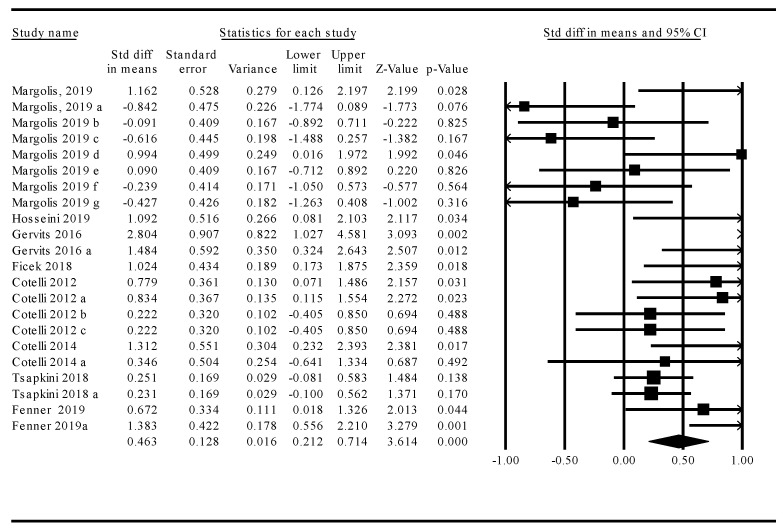
Overall meta-analysis effect sizes of transcranial direct current stimulation (tDCS) and transcranial magnetic stimulation (TMS) treatment studies in primary progressive aphasia (PPA). Corresponding forest plots demonstrate the overall treatment effects (<0 favors sham; >0 favors active stimulation).

**Figure 3 brainsci-10-00597-f003:**
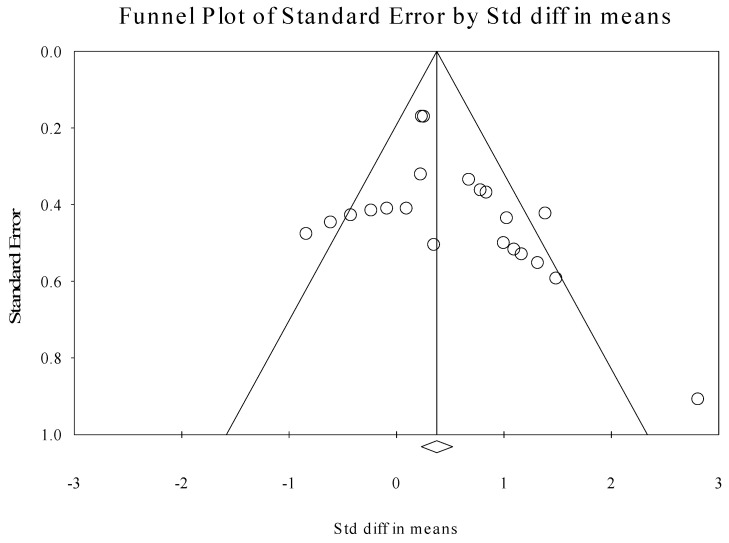
Funnel plot displaying tDCS and TMS effects for the assessment of publication bias.

**Table 1 brainsci-10-00597-t001:** Studies included in the meta-analysis.

Study	Modality	Study Design	Sample Size	PPA Variant	Sessions	Stimulation Parameters	Location of Stimulation	Concurrent Task	Primary Outcome Measure
[25]	tDCS	Between-subject study	16	naPPA	10	2 mA for 25 min (30 s ramp up/down)	Anode: Left DLPFC (BA8/9); cathode: Right arm	Individualized speech therapy	Aachener Aphasie Test (AAT) naming subtest: mean correct response
[19]	tDCS	Open label pilot study	6	naPPA/lvPPA	10	1.5 mA for 20 min (30 s ramp up/down)	Anode: Left fronto-temporal region (F7); cathode: Left occipito-parietal region (O1)	Narration of wordless picture books	Cookie theft picture task: elicited speech production from the Boston Naming Test; Language test for the reception of grammar (L-TROG): mean correct response
[61]	tDCS	Within-subject crossover	24	naPPA/lvPPA/svPPA	15	2 mA tDCS for 20 min (active); 30 s (sham)	Anode: Left frontal lobe (F7); cathode: right cheek	Oral and written picture naming spelling therapy	Naming/spelling accuracy: percentage of correct response/correct letters
[28]	tDCS	Within-subject crossover	36	naPPA/lvPPA/svPPA	15	2 mA for 20 min (30 s ramp up/down)	Anode: Left frontal lobe (F7); cathode: Right cheek	Written naming spelling therapy	Naming spelling accuracy: percentage of correct letters from treated versus untreated words
[62]	tDCS	Within-subject crossover	11	naPPA/lvPPA	15	2 mA for 20 min (30 s ramp up/down)	Anode: Left frontal lobe (F7); cathode: Right cheek	Oral and written naming spelling treatment	Written naming performance: mean letter accuracy
[27]	tDCS	Within-subject crossover	6	naPPA	10	1.5 mA for 20 min (30 s ramp up/down)	Anode: Left prefrontal region; cathode: Left occipital region	Category fluency	Category fluency: mean number of words
[18]	rTMS	Within-subject crossover	10	naPPA	1	rTMS: 20 Hz frequency at 90% MT; 500 ms; 84 trains	Left and Right DLPFC	Action/object naming	Action and object naming: mean correct response
[26]	rTMS	Within-subject crossover	6	naPPA	2	rTMS: 20 Hz frequency at 90% MT; 1000 ms; 84 trains	Left and Right DLPFC	Action/object naming task and Stroop task	Action and object naming: mean correct response; Stroop color-word accuracy

PPA: primary progressive aphasia; naPPA: nonfluent/agrammatic PPA; lvPPA: logopenic variant PPA; svPPA: semantic variant PPA; tDCS: transcranial direct current stimulation; rTMS: repetitive transcranial magnetic stimulation; MT: motor threshold; DLPFC: dorsolateral prefrontal cortex.

**Table 2 brainsci-10-00597-t002:** Study sample demographics.

Study	Stimulation Modality	Sample Size	Mean Age	Education (Mean Year)	Percent Male	Disease Duration (Mean Year)
[25]	tDCS	16	63.4	9.3	36	2
[19]	tDCS	6	66.2	16.3	17	4.2
[61]	tDCS	24	67.2	-	54	4.9
[28]	tDCS	36	67.9	16.3	55	5.8
[62]	tDCS	11	67.6	-	64	5
[27]	tDCS	3	67.0	14.3	33	4.8
[18]	rTMS	10	69.1	-	20	2.3
[26]	rTMS	6	67.0	15	67	-

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
