# Peer review of "Efficacy of Noninvasive Brain Stimulation (tDCS or TMS) Paired with Language Therapy in the Treatment of Primary Progressive Aphasia: An Exploratory Meta-Analysis"

_brainsci, 2020, doi:10.3390/brainsci10090597_

Round 1
Reviewer 1 Report
This is a well-designed and well-written systematic review, on a relevant topic. Critically, given the variability in study parameters employed alongside the increasing number of studies in this field, a review of this kind is timely. The data extraction, coding and analysis procedures are accurate and described in sufficient detail. However, certain conceptual aspects of relevance should be taken into account. My main concern is the lack of detail provided and accounted for relating to the behavioral treatment task employed alongside neuromodulation. Having said that, I believe this research provides a worthwhile contribution to the literature, and can be revised to account for those details.
Introduction
Page 2, line 48 – cite de Aguiar et al. (2015) for post-stroke aphasia.
Page 2 – it would be relevant to also discuss here stimulation sites used in previous studies, and their relations to behavioral tasks employed.
Page 3 – differences in spatial resolution may also be relevant to pairing of the stimulation site to a behavioral task. This should be mentioned as an advantage of TMS, the importance of spatial resolution in clinical applications may not be as relevant as it is for research applications.
Method
Page 5, line 182 should read “inclusion and exclusion criteria ARE” (not is).
Page 5, Little detail is provided concerning thew concurrent task. For instance, when unspecified, do the treatment tasks use objects or actions? What is the nature of the stimuli presented (pictures/videos)? What is the delivery mode (computerized/by SLT), what kind of feedback and cues were given when the participants failed to retrieve the targets (type of cue and cueing hierarchy)? What was the frequency and duration of therapy sessions? Given the hypothesized central role of the concurrent behavioral therapy, these variables should be taken into account. At the moment, only the number of sessions is considered in statistical analysis. However, this does not say a lot about the content of therapy.
Results
No differences between treated and untreated items – does this mean generalization or regression to the mean?
Page 11, line 310 – this is an odd sentence “…in the tDCS and TMS for PPA literature”.
Discussion
Page 12, line 371 – Please edit “we had had expected…”.
It will also be important to discuss here the noted heterogeneity between studies. As samples are generally small, this can be due to sample selection differences. However, a likely source of heterogeneity between studies is also the difference in the features of the behavioral treatment employed.
Author Response
Response to Reviewer 1 Comments: Dear reviewer, thank you for taking the time to review the manuscript and your helpful comments.
Point 1: Introduction Page 2, line 48 – cite de Aguiar et al. (2015) for post-stroke aphasia.
Response 1: This reference has been added.
Point 2: Page 2 – it would be relevant to also discuss here stimulation sites used in previous studies, and their relations to behavioural tasks employed.
Response 2: We have added text (page 2, lines 91-103) regarding stimulation sites and the relation to the behavioral task employed in the context prior research in PPA.
Point 3: Page 3 – differences in spatial resolution may also be relevant to pairing of the stimulation site to a behavioral task. This should be mentioned as an advantage of TMS, the importance of spatial resolution in clinical applications may not be as relevant as it is for research applications.
Response 3: We agree with this comment and have added text (page 3, lines 125-128) to address the relevance of spatial resolution in the pairing of stimulation and behavioral task.
Point 4: Page 5, Little detail is provided concerning the concurrent task. For instance, when unspecified, do the treatment tasks use objects or actions? What is the nature of the stimuli presented (pictures/videos)? What is the delivery mode (computerized/by SLT), what kind of feedback and cues were given when the participants failed to retrieve the targets (type of cue and cueing hierarchy)? What was the frequency and duration of therapy sessions? Given the hypothesized central role of the concurrent behavioral therapy, these variables should be taken into account. At the moment, only the number of sessions is considered in statistical analysis. However, this does not say a lot about the content of therapy.
Response 4: We acknowledge that differences between behavioral therapy approaches likely impact the response to neuromodulation interventions. However, we ultimately felt that a detailed evaluation of behavioral language treatment approaches fell beyond the intended scope of this manuscript, in which we aimed to identify whether stimulation improved language outcomes across behavioral interventions. We therefore used liberal criteria for types of language therapies and included a wide range of different therapy approaches. In light of this, the differences between behavioral therapies could not be readily assessed systematically. We have added additional text (page 5, lines 190-193) to explain that the behavioral therapy approach was included irrespective of the type/form of language therapy. Details regarding the concurrent task employed are provided in Table 2, alongside the specific outcome measures that were assessed, and the frequency of sessions. Duration of therapy could not be evaluated and was not included in the analysis. Finally, we also now acknowledge in our revised discussion section that the heterogeneity of the behavioral therapy approaches was a limitation of the manuscript.
Point 5: Results – no differences between treated and untreated items – does this mean generalization or regression to the mean?
Response 5: We interpret this comment from the reviewer as possibly expressing some degree of surprise at the fact that our analysis yielded no significant difference between treated and untreated items. This is understandable, as this result ran counter to our expectations as well. In our discussion section, we make the conservative suggestion that the finding “may indicate that practice effects were not particularly strong in the included studies.” However, the reviewer raises at least two other possibilities for why there may have been no difference between treated and untreated items. We agree that generalization of improvements in language ability comprises one potential explanation (and the most optimal outcome), so we have modified our discussion section (page 14, line 409-410) to include this possibility. By contrast, we did not feel that the results were consistent with regression to the mean because most of the studies that informed our analysis featured a sham control condition. It seemed unlikely to us that there would be regression to mean performance across multiple studies that impacted performance in the tDCS condition but not the sham condition. Therefore we did not include this potential interpretation of the results in our revised discussion section.
Point 6: Page 11, line 310 – this is an odd sentence “…in the tDCS and TMS for PPA literature”.
Response 6: The text has been revised to read “…in this sample of literature”
Point 7: Discussion – Page 12, line 371 – Please edit “we had had expected…” It will also be important to discuss here the noted heterogeneity between studies. As samples are generally small, this can be due to sample selection differences. However, a likely source of heterogeneity between studies is also the difference in the features of the behavioral treatment employed.
Response 7: Thank you for noting the typo. It has been fixed. We agree with your comment and have added text (page 15, lines 445-449) highlighting the heterogeneity between studies due to the inclusion of different behavioral treatments and have noted this as a limitation in our revised discussion section.
Reviewer 2 Report
Dear Authors;
Thankyou for this interesting review. The manuscript is very clearly written and well structured. The methods appear thorough and well executed. Discussion and Conclusions are optimistic but appropriately highlight the limitations of the review and its outcomes related to small number of studies, hetergeneous effects etc. The study will make a valuable contribution for future research directions.
I have just some minor comments:
line 182: "criteria" is a plural noun" - cahnge "Inclusion and exclusion criteria is summarized" to "... are summarized"
Table 1 may not be necessary as it duplicates information in the text as well as in Figure 1.
It is difficult to follow the numbers in Figure 1. There are 77 after duplicates. The next level is confusing - 58 screened + 19 excluded at screening adds to 77 but is it that there were 58 that survived screening and, of those, 19 were excluded from full-text assessment to give 39 for full assessment (58-19=39)?
Author Response
Response to Reviewer 2 Comments: Dear reviewer, thank you for taking the time to review the manuscript and your helpful comments.
Point 1: Line 182: “criteria” is a plural noun – change “inclusion and exclusion criteria is summarized” to “…are summarized. 

Response 1: Thank you, the text has been corrected.
Point 2: Table 1 may not be necessary as it duplicates information in the text as well as in Figure 1.
Response 2: We agree that content in Table 1 is redundant with information in text and have removed it from the manuscript.
Point 3: It is difficult to follow the numbers in Figure 1. There are 77 after duplicates. The next level is confusing – 58 screened + 19 excluded adds to 77 but is it that there were 58 that survived screening and, of those, 19 were excluded from full-text assessment to give 39 for full assessment (58-19=39).
Response 3: Thank you for this helpful comment. There was an error in the number of records screened and excluded, so we have revised the flow chart. 77 articles were screened after duplicates were removed. Of those, 38 were excluded as off topic articles that did not permit full-text assessment for eligibility. The remaining 39 articles were fully assessed for eligibility, and the subsequent numbers in the figure remain unchanged.